# The Dark Side of Healthy Eating: Links between Orthorexic Eating and Mental Health

**DOI:** 10.3390/nu12123662

**Published:** 2020-11-28

**Authors:** Jana Strahler

**Affiliations:** 1Department of Psychotherapy and Systems Neuroscience, Psychology and Sport Sciences, Justus Liebig University Giessen, Otto-Behaghel-Str. 10H, 35304 Giessen, Germany; jana.strahler@psychol.uni-giessen.de or jana.strahler@gmail.com; Tel.: +49-641-26332; 2Department of Health Psychology and Applied Diagnostics, Human and Social Sciences, University of Wuppertal, 42119 Wuppertal, Germany

**Keywords:** healthy eating, orthorexia nervosa, psychopathology, wellbeing

## Abstract

Orthorexia nervosa (OrNe) describes a behavior where eating overly healthy develops into an obsession leading to significant impairment and stress. Initial studies support a bi-dimensional structure of orthorexic eating with one dimension healthy orthorexia (HeOr, interest in healthy eating), which can be distinguished from the dimension OrNe. The present study pursued the goals to examine the negative consequences of OrNe on mental health, whether HeOr buffers these effects, and the role of gender. Data from two cross-sectional online surveys were combined (study 1 *n* = 385, 310 women; study 2 *n* = 398, 265 women; mean age: 28.9 ± 12.0 year) both generating data on psychological wellbeing, life satisfaction, stress, anxiety, and depressive symptoms in relation to OrNe and HeOr (Teruel Orthorexia Scale). By means of correlation and moderation analyses, OrNe was shown to be associated with poorer mental health, especially in the female sample. In terms of HeOr, clear gender differences appeared. There were no meaningful correlations in women. In men, however, HeOr correlated with better mental health. Further, the link between OrNe and poorer mental health was mitigated when there were high HeOr levels. Present findings support the hypotheses that OrNe is associated with pathological consequences and that HeOr may act as a buffer for these consequences. Gender differences in the clinical manifestation of orthorexic eating confirm previous knowledge and have important implications for targeted prevention and treatment strategies.

## 1. Introduction

Proper nutrition and a balanced diet are essential in maintaining physical and mental health, and wellbeing [1]. While a healthy diet is protective, an unhealthy diet constitutes a major risk factor for anxiety and depression across all age groups [2,3]. The reverse side of the focus on healthy eating, however, is when eating overly healthy develops into an obsession, a behavior labeled orthorexic eating or in its extreme form Orthorexia nervosa (OrNe) [4]. OrNe includes multiple symptoms such as an fixation on the nutritional quality of food, disturbed psychological functioning, restrictive eating despite medical contraindications and consequences, and interference with social life and relationships [5]. While orthorexic eating tendencies are spread across genders, age groups, and cultures [6,7], OrNe is not formally recognized as a distinct disorder in medical and psychiatric classification systems. Missing formal diagnostic criteria, lack of empirical data on prevalence, psychopathology, clinical consequences, and treatment as well as the differentiation from other diagnostics categories contribute to the controversy over whether OrNe should be a discrete condition. Still, research provides sufficient anecdotal evidence that when healthy eating turns into OrNe, it leads to significant impairment and stress [8]. Some case reports describe the negative health effects of very extreme forms of orthorexic eating, including malnourishment, severe underweight, and cardiovascular and endocrine disturbances [9]. Further empirical studies, even though mainly cross-sectional and conducted in non-clinical settings, provide evidence for the negative impact of OrNe on both psychological and physical wellbeing [10]. Recently, there has been a growing effort to employ qualitative research methods. These studies provide insights into most relevant self-perceived drivers of healthy eating, contributors to the development of a preoccupation with healthy eating and OrNe, the conditions progress and treatment [11,12]. This research also supports the theoretical model of a bi-dimensional structure of orthorexic eating [13]. Although one may begin a healthy diet to improve health and wellbeing, the focus on eating only healthy foods can turn into an obsession and fixation with detrimental effects on physical, psychological, and social wellbeing. Initial studies provide support for this bi-dimensionality. While OrNe, the pathological preoccupation with healthy diet, relates to higher negative affect, the interest in healthy eating, termed healthy orthorexia (HeOr), was predictive of heightened positive affect [14]. In addition, motives for food choices differed between these dimensions. Weight control and affect regulation were the main predictors of OrNe, while for HeOr this was health content [15]. Overall, there is a clear need for further research about the negative consequences of orthorexic eating under consideration of the distinct dimensions of eating (overly) healthy. Such evidence would then map the pathological relevance of such eating behaviors and help to identify persons in need of treatment.

To address this, we combined data from two different studies, both adopting a cross-sectional survey-design and both generating data on psychological wellbeing, life satisfaction, stress, and anxious and depressive symptoms. As OrNe should be differentiated from non-pathological healthy eating by the clinically significant distress or impairment in functioning it produces, HeOr was expected to positively relate to wellbeing measures and negatively relate to strain measures; OrNe should show the opposite pattern. In addition, we also expected the level of HeOr to moderate the relationship between OrNe and mental health. From previous research on the link between HeOr and positive affect [14] and health advantages of a eating healthy, the interest in healthy eating may act as a buffer for the negative consequences of OrNe. To take account of gender differences in orthorexic eating and self-reported mental health and wellbeing [16,17], the moderating effects of gender were expected and examined.

## 2. Materials and Methods

### 2.1. Participants and Design

Data for this analysis stems from two different online surveys, both recruiting a convenience sample from the general population. The surveys took place successively and different study objectives were advertised. Study 1 was described as a study of the association between health-conscious behavior and person characteristics (conducted between April and July, 2019); study 2 was about the link between (competitive) sports and health (conducted between August and October, 2019). While we cannot exclude the possibility that one person participated in both studies, the risk seems to have been low. Comparing the combined information on gender, age, height, educational level, and eating style showed two very similar data sets. Both sets were excluded from further analyses. The link to the survey (SosciSurvey.com) was shared via university mailing lists and social media. Participation was voluntary, a study agreement was presented on the first page of the survey, and subjects were required to click an acceptance box containing informed consent. The Ethics Committee of the University of Wuppertal approved both studies (study 1 reference MS/BBL 190411, study 2 reference MS/BBL 190718). (Parts of the data of study 1 have already been published [18]. None of the analyses presented herein overlap with this previous publication.)

Inclusion criteria were being aged 16 years and older, and ascribing oneself clearly to the male or female gender. Initially, 800 complete data sets (study 1: 391, study 2: 409) were available. We excluded the *n* = 2 very similar data sets, *n* = 2 subjects indicating ‘diverse’ as their gender, and *n* = 12 subjects completing the survey in under 10 min or being two times faster than the median of all participants. We also asked for current extreme stressors (e.g., bereavement). This resulted in the exclusion of one other dataset leaving us with 783 valid datasets for analyses (study 1: 385, study 2: 398). See Table 1 for demographic characteristics of the study sample.

### 2.2. Survey Measures

Orthorexic eating traits were examined by means of the Teruel Orthorexia Scale (TOS) [13], which allows the acquisition of the proposed bi-dimensional structure of orthorexic eating on the two subscales—healthy orthorexia (HeOr, 9 items) and orthorexia nervosa (OrNe, 8 items). Items are rated on 4-point Likert scales (completely disagree to completely agree). Item ratings are summed up with higher scores indicating higher levels on the subscales. Internal consistencies of both subscales were high in the present study (see Table 2).

The WHO-5 Well-being Index (WHO-5) [19] and the Life Satisfaction Scale (L1) [20] were employed to measure psychological well-being and life satisfaction, respectively. With the WHO-5, respondents rate how well each of the five statements apply with reference to the last two weeks and score from 5 (all of the time) to 0 (none of the time). Scores are summed up and multiplied by 4 to form a percentage scale from 0 (absence of well-being) to 100 (maximal well-being). Psychometric properties of this short questionnaire are adequate [21]. The single-item measure L1 reads “All things considered, how satisfied are you with your life these days?” and answers are recorded with an 11-point scale from 0 (not at all satisfied) to 10 (completely satisfied). The L1 showed moderate re-test stability, adequate construct validity, and results support the L1 as measure of the cognitive evaluation of one’s own quality of life. The reliability coefficient of the WHO-5 herein was high (see Table 2).

Perceived stress, i.e., perceiving one’s life as unpredictable, uncontrollable, and overloaded, was assessed with the Perceived Stress Scale (PSS-10) [22]. Responses are given on 5-point scales from 0 (never) to 4 (very often) and sum scores are created. Internal consistency was high in the present study (see Table 2).

To measure emotional distress, two different scales were used within the two studies, the 21-item Depression, Anxiety and Stress Scale (DASS21) [23] and the 14-item Hospital Anxiety and Depression Scale (HADS) [24]. Both questionnaires are reliable and valid screening instruments for depressive and anxiety symptoms and internal consistencies were in acceptable and good ranges in the present datasets (see Table 2).

An accompanying questionnaire gathered demographic information on age, gender, weight, and height to calculate body mass index (BMI), school education (none or still in school, primary, secondary, university-entrance), marital status (un-married, married), and eating style (vegan, any form of vegetarian, rare meat, no restrictions). For BMI, categories were created according to current recommendations (below 18.5 = underweight, 18.5 to 24.9 = normal weight, 25.0 to 29.9 = overweight, >30.0 = adiposity) [25].

### 2.3. Statistical Procedures

First, factor structure of the items on the TOS was extracted using principal component analysis with oblimin rotation (allowing the factors to be correlated) and the scree plot. Confirmatory factor analysis with a maximum likelihood estimator compared the fit with the two proposed theoretical factors. Model fit was assessed with the Comparative Fit Index (CFI) and the root mean square residual (RMSEA), with CFI values at or above 0.90 and RMSEA values at or below 0.08 representing cutoffs for an acceptable model fit [26].

The Shapiro Wilk test indicated that the data were not normally distributed but Q-Q plots indicated no or only mild skewness of the data. Given the fairly large sample size and because of the central limit theorem, parametric statistical tests are assumed to be robust to mild non-normality [27]. Hence, Student’s *t* test and X^2^ statistic compared means and frequencies, respectively. The employed effects sizes were Hedges’ g to account for unequal sample sizes, and Cramer’s V, respectively.

Correlations between study variables were examined using Pearson’s correlation. Of note, only r > 0.2 will be interpreted as this is the recommended minimum effect size representing a “practically” significant effect [28]. Fisher’s z indicated differences between correlations among the male and female sample. Cohen’s q is used as effect size to interpret z.

Hierarchical multiple regressions examined if the different dimensions of orthorexic eating (HeOr and OrNe) contribute to mental health (WHO-5, L1, PSS-10; for study 1: HADS; for study 2: DASS21). Main effects of gender were accounted for in block 1, and block 3 included the HeOr × OrNe interaction to examine whether the association between OrNe and mental health depends on the level of HeOr. The predictors were mean centered, to avoid potentially high multicollinearity with the interaction term, and the interaction term between OrNe and HeOr was created [29]. All assumptions were met (correlation between predictors, all r < 0.578; VIF < 1.6; Cook’s distance < 0.1 except one data point for HADS-D and DASS21-D; normally distributed residuals as indicated from normal P-P-plots). To illustrate possible HeOr × OrNe interaction effects, four groups based on median split (higher/lower HeOr, higher/lower ON) were created and plotted.

IBM SPSS Statistics (v.23.0 for Mac, IBM Corp., Armonk, N.Y., USA) and AMOS (v.27) were used for statistical analyses. *p* values < 0.05 are considered significant.

## 3. Results

### 3.1. Demographic Characteristics

As is to be expected from our recruitment strategy, the sample was rather young (M = 28.89, SD = 11.99; Md = 24.00), unmarried, and highly educated. More than 70% of the sample was of normal weight and about 60% reported to follow an omnivorous eating style. Please see Table 1 for the details. Comparing the male and female samples showed that men were older (t_age_(277.5) = 5.77, *p* < 0.001, g = −0.559; M_men_ 33.67 ± 15.07, M_women_ 27.16 ± 10.13) and had a slightly higher BMI (t_BMI_(781) = 5.36, *p* < 0.001, g = −0.434; X^2^_BMI_ = 28.438, *p* < 0.001, V = 0.191; M_men_ 24.25 ± 4.06, M_women_ 22.45 ± 4.08). There was no difference in educational attainment (X^2^_education_ = 1.470, *p* = 0.689, V = 0.043) but men were more likely married (X^2^_marital status_ = 7.611, *p* = 0.006, V = 0.099; men 22.1%, women 13.9%) and indicated more often that they eat an omnivorous diet (X^2^_eating style_ = 30.199, *p* < 0.001, V = 0.196; men 74.0%, women 52.5%). Demographic variables, except gender, will therefore not be considered in the following analyses to avoid multicollinearity and the inclusion of variables that supply redundant information.

### 3.2. Orthorexic Eating Differentially Linked to Wellbeing and Strain Measures

A principal component analysis and the scree plot confirmed the basic two-factor structure of the TOS. The pattern of item loadings confirmed item-scale allocations. Exception was item 7 “I’d rather eat a healthy food that is not very tasty than a good tasting food that isn’t healthy” which showed a cross-load > |0.3| on both scales. As the cross-load was higher for the HeOr subscale (*r* = 0.46, OrNe *r* = 0.34) and to not affect content validity of the measure, this item was retained. Confirmatory factor analysis indicated a data-to-proposed-model fit slightly above the cutoffs (X^2^ (118) = 792.3, *p* < 0.001, CFI = 0.873, RMSEA = 0.085 with CI of 0.080 to 0.091). Of note, TOS levels were comparable between study samples (TOS-HeOr: M = 12.12/11.66, *p* = 0.207, α_Cronbach_ = 0.84/0.84; TOS-OrNe: M = 3.34/3.38, *p* = 0.875, α_Cronbach_ = 0.85/0.89 [study 1/study 2]).

Table 2 provides descriptives of the study variables for the total sample and separately for the male and female sample. Neither of the two subscales of orthorexic eating differed between genders (t_HeOr_(781) = −1.70, *p* = 0.089; t_OrNe_(468.5) = −1.85, *p* = 0.064) but there were some differences in terms of wellbeing and strain measures. The difference in psychological wellbeing was not significant, but men reported slightly higher levels (t_WHO-5_(781) = −1.70, *p* = 0.089, g = −0.252). Stress was significantly lower in men (t_PSS-10_(781) = −5.70, *p* < 0.001, g = 0.461; t_DASS21-S_(396) = −4.06, *p* < 0.001, g = 0.432), as were anxiety and depressive symptoms, but only in the study 2 subsample (employing the DASS21) and with small effect sizes (t_HADS-A_(383) = −1.23, *p* = 0.218; t_DASS21-A_(353.9) = −3.03, *p* = 0.003, g = 0.287; t_HADS-D_(95.0) = 1.65, *p* = 0.103; t_DASS21-D_(396) = −1.98, *p* = 0.049, g = 0.209).

HeOr and OrNe correlated significantly (*r* = 0.488, *p* < 0.001) with comparable correlation scores between men and women (r_men_ = 0.493, r_women_ = 0.491, both *p* < 0.001, z = -0.032, *p* = 0.487). As hypothesized, OrNe was positively associated with all strain measures; the effects were moderate. Associations with wellbeing and life satisfaction appeared negative and small-to-medium sized (see Table 3). Analyses separated by gender indicated that the associations were mainly driven by the female sample. In men, only anxiety as measured by the DASS21 appeared significantly correlated with OrNe and the strength of the association was comparable to the score in women (z = −0.032, *p* = 0.487). In contrast to our expectations, HeOr was hardly related to mental health. The only meaningful association (i.e., r > 0.2) was a positive correlation with the WHO-5 index and this correlation appeared higher in men as compared to women (z = −1.69, *p* = 0.046, q = 0.137). Gender-separated analyses mirrored total sample findings for the female subgroup while in men, HeOr was positively correlated with wellbeing and life satisfaction, and negatively correlated with stress (PSS-10) and depressive symptoms (DASS21-D).

### 3.3. Healthy Orthorexia Moderates the Consequences of Orthorexia Nervosa

Hierarchical multiple regressions were calculated to predict mental health based on HeOr, OrNe, and the HeOr × OrNe interaction (see Appendix A for the details). (Sub-study analyses confirmed the pattern of findings. Exception was a missing HeOr*OrNe effect on PSS-10 in study 1, β = −0.10, p = 0.076 (study 2: β = −0.15, p = 0.007). Explained variance was also similar (study 1/study 2: WHO-5 14.1%/15.1%, L1 12.7%/10.0%, PSS-10 18.5%/16.3%).) Recoding the HeOr and OrNe scale into a high versus low group created a grouping variable that should picture the HeOr × OrNe interaction.

For WHO-5, the results of the full model indicated the predictors explained 14.6% of the variance (F(4778) = 34.42, *p* < 0.001). Gender was a significant contributor in the first block (β = −0.11, *p* = 0.002). In block 2, it was found that OrNe significantly predicted wellbeing (β = −0.34, *p* < 0.001), as did HeOr (β = 0.38, *p* < 0.001). Including the HeOr × OrNe interaction (β = 0.11, *p* = 0.009) accounted for a small but significant proportion of variance in wellbeing (∆R^2^ = 0.8%, F_change_(1778) = 6.93, *p* = 0.009). Similarly, the L1 full model showed that the predictors were able to account for 10.3% of the variance in life satisfaction (F(4778) = 23.49, *p* < 0.001). Gender was not a significant predictor (β = −0.05, *p* = 0.127), but OrNe (β = −0.33, *p* < 0.001) and HeOr (β = 0.26, *p* < 0.001) were. The HeOr × OrNe interaction further contributed to explained variance in life satisfaction (β = 0.13; ∆R^2^ = 1.2%, F_change_(1778) = 10.06, *p* = 0.002). Subjects showing higher TOS-OrNe and at the same point lower TOS-HeOr showed lowest wellbeing and life satisfaction (see Figure 1A,B).

The full predictor model was able to account for 17.8% of the variance in perceived stress as measured by the PSS-10 (F(4, 778) = 43.21, *p* < 0.001). All predictors appeared significant (β_sex_ = 0.20, *p* < 0.001; β_OrNe_ = 0.41, *p* < 0.001; β_HeOr_ = −0.28, *p* < 0.001; β_interaction_ = −0.11, *p* = 0.006) with the two orthorexia dimensions contributing the most explained variance (∆R^2^ = 13.4%) and only little contribution by the HeOr × OrNe interaction (∆R^2^ = 0.8%). Stress levels were most pronounced in those showing higher TOS-OrNe and at the same point lower TOS-HeOr (see Figure 1C). The model including the DASS21-S as dependent variable mirror these findings, except a missing contribution of the HeOr × OrNe interaction (∆R^2^ = 0.5%, β = −0.09, *p* = 0.124).

In regard to depressive symptoms, regression models appeared quite similar for both scales employed. For either tool, the full model explained 17.2% of the variance. The contribution of gender was only marginal (both ∆R^2^ = 1.0%, β_HADS-D_ = −0.10, *p* = 0.051; β_DASS21-D_ = 0.10, *p* = 0.049). OrNe and HeOr differentially predicted depressive symptoms (OrNe: β_HADS-D_ = 0.41, β_DASS21-D_ = 0.45; HeOr: β_HADS-D_ = −0.35, β_DASS21-D_ = −0.25, all *p* < 0.001), and the HeOr × OrNe interaction further contributed a small proportion of variance in depressive symptoms (β_HADS-D_ = 0.41, *p* < 0.001 ∆R^2^ = 2.7; β_DASS21-D_ = 0.45, *p* < 0.001, ∆R^2^ = 0.9). Highest depressive symptoms were reported by subjects showing higher TOS-OrNe and lower TOS-HeOr levels (see Figure 1D).

Explained variance in anxiety symptoms was different between the tools. While the full predictor model accounted for 18.3% in HADS-A scores (F(4, 380) = 22.44, *p* < 0.001), this was 12.8% for the DASS-S subscale (F(4, 393) = 15.54, *p* < 0.001). As for depression, the individual contribution of gender was only small (β_HADS-A_ = 0.06, *p* = 0.218, ∆R^2^ = 0.4%; β_DASS21-A_ = 0.14, *p* = 0.007, ∆R^2^ = 1.8%). OrNe and HeOr appeared as significant predictors (OrNe: β_HADS-A_ = 0.50, β_DASS21-A_ = 0.38; HeOr: β_HADS-A_ = −0.23, β_DASS21-A_ = −0.15, all *p* < 0.001) but there was no additional value of the HeOr × OrNe interaction (β_HADS-A_ = −0.05, *p* = 0.358 ∆R^2^ = 0.2; β_DASS21-A_ = −0.08, *p* = 0.192, ∆R^2^ = 0.4, not shown).

## 4. Discussion

### 4.1. Summary of Main Findings

This study pursued the goal to better understand the pathological consequences of orthorexia nervosa and to validate the theoretical assumption of a bi-dimensional structure of orthorexic eating. We assumed that the interest in healthy eating (1) differentiates from orthorexia nervosa in their relation to mental health, and (2) buffers negative consequences of this pathology. Indeed, the two proposed sub-dimensions of orthorexic eating, HeOr and OrNe [13], were differentially related to wellbeing (psychological wellbeing, life satisfaction) and strain measures (stress, anxiety, depression). While OrNe, the pathological obsession and fixation on healthy eating was clearly associated with higher strain and worse wellbeing, HeOr, the healthy interest in proper diet, was unrelated to strain measures but positively associated with wellbeing. Effects appeared small-to-medium sized. Levels of HeOr moderated the relationship between OrNe and mental health with least favorable levels in subjects reporting on higher OrNe but lower HeOr, and most favorable levels in subjects reporting on lower OrNe but higher HeOr. Examining gender as a contributing factor indicated that findings in regard to OrNe were mainly driven by the female sample. In contrast, HeOr was more clearly linked to better mental health and wellbeing in the male sample.

### 4.2. Pathological Consequences of Orthorexia Nervosa

From previous cross-sectional, case-report, and interview studies, OrNe was hypothesized to be related to a significant reduction in psychological wellbeing, heightened impairment, and higher stress [8,10,11,12]. The present data showed that this was mainly true for strain measures (here stress, anxiety, and depressive symptoms) but not or less so for wellbeing and life satisfaction. Higher strain may thus not necessarily be reflected in worse wellbeing in those affected by OrNe. This distinction also fits well within frameworks in which mental health is understood as a multidimensional construct where wellbeing and ill-being are closely linked but independent dimensions constituting mental health [30,31]. One factor that has contributed to the present results is gender. Total sample findings were driven by the female sample while in men the only significant correlation of OrNe was with anxiety. Having examined >200 men, it seems unlikely that low statistical power contributed to this null finding in the male subsample. Gender differences in the clinical manifestation of OrNe have been highlighted in previous studies. Women with orthorexic eating were more likely to report positive feelings about their eating, while orthorexic men focused more on normative behaviors and reported more problems from this rigid eating behavior [32,33]. This heterogeneity in the clinical manifestation of OrNe may be attributed to biological and sociocultural factors such as attribution of gender roles. If these clinical variations hold true, findings have important implications for targeted prevention and treatment strategies. Gender differences in orthorexic characteristics also emphasize the need to examine OrNe as a multidimensional construct and how gender differentially impacts the different dimensions of dysfunction.

### 4.3. Healthy Orthorexia Acts as a Buffer of Orthorexia Nervosa

Our hypotheses–that HeOr relates to better mental health–was only partially supported. Data showed that HeOr was correlated with higher psychological wellbeing but neither with life satisfaction nor any of the strain measures. Although previous research has shown HeOr to predict heightened positive affect [14], this clarity was missing in our total sample analyses. Again, gender differences need to be discussed. In fact, there were no meaningful correlations in women. In men, however, HeOr correlated with higher wellbeing and life satisfaction as well as lower stress and depressive symptoms. At first sight, this finding is in contrast to findings according to which dietary choices impact women’s wellbeing more than men’s [34]. Generally, women have a better nutritional knowledge and higher awareness, and they confer a greater importance to healthy eating [35]. These tendencies, however, may make women as compared to men also more concerned about healthy eating [36] and their own dietary behaviors. This ambivalence is discussed as an underlying factor in women’s restrained and disordered eating behaviors [37] and may have also contributed to the varying correlations between HeOr and mental health shown herein. As we have no information on actual food consumption and preferences, this however remains speculative. In addition, an important next step for future research is to better understand motives for food choices, such as health beliefs versus weight control, which vary between men and women [35]. Besides these behavioral and psychological factors, there is also evidence for gender-specific brain activation patterns underlying food choices [38]. Whether neural food cue reactivity modulates orthorexic eating tendencies is currently unknown.

Results of the moderation analyses confirmed our assumption that levels of HeOr may act as a buffer for the negative consequences of OrNe. Subjects scoring high on the OrNe dimension showed poorer mental health but this link was mitigated when there were also high HeOr levels. This finding suggests a close relationship between the dimensions—in fact, they correlated by 0.5—and a more complete understanding of their commonalities and differences will inform prevention and treatment strategies. For example, food choice motives differed between the dimensions; HeOr tendencies related to health content, OrNe was best predicted by weight control and affect regulation [15]. Such knowledge helps clinicians to draw conclusions for targeted interventions. Presumably, cognitive-behavioral interventions focusing on the food-related cognitions and beliefs in regard to “healthy” eating can be derived from this knowledge.

### 4.4. Critical Reflections

Though this study has many strengths (sample size, valid measurement tools, no missing data due to programming), some potential limitations must be considered. The majority of correlations between OrNe and mental health identified were of small to medium effect size, particularly in men, and may therefore be of limited clinical importance. For women, associations with strain measures were of comparably larger magnitude and may be clinically more relevant. As for correlations, explained variance in regression models (R^2^) was only small. This needs to be considered when evaluating the models. However, coefficient estimates were significantly different from zero, the models performed equally well on data from study 1 and study 2 (cross-validation, see footnote 2), and R^2^ values as well as low-to-medium sized correlations mirror previous study’s results [39,40]. Overall, orthorexic eating slightly contributes to wellbeing and strain but much of the variance remains unexplained, indicating the need for further research on additionally important factors. The study was based on data from two different research trials which, however, took place in close proximity to each other. Thus, there is a possibility that one person has participated in both studies. This possibility was considered and investigated via comparing information on gender, age, height, educational level, and eating style. The only two exactly matching data sets were excluded from analyses. Our recruitment strategy attracted a rather young, female, well-educated sample and it remains unclear how results generalize to socio-culturally more diverse samples. While we employed a valid tool to examine the bi-dimensional nature of orthorexic eating, this measure neither provides cut-offs beyond which the behavior can be considered pathological nor does it replace a formal clinical diagnosis. For this purpose, clinical interviews are necessary, which also ask for reasons for following this dietary style and investigate other disordered eating habits. As the presence of any current or past eating disorder has not been evaluated, investigating possible overlaps with OrNe and disentangling impairments related to OrNe as compared to eating disorder symptoms [39,40] was not possible. Finally, cross-sectional studies offer a snapshot of a single moment in time; they do not qualify for making causal inferences. Observational longitudinal studies will have to increase our knowledge of a possible sequence of orthorexic eating behaviors, clarify risk factors for when healthy eating turns into a pathological obsession, and examine the effectiveness of preventive and therapeutic interventions.

## 5. Conclusions

In sum, the present study identified pathological consequences of orthorexic eating habits, with small to medium effect sizes. Men and women were affected to varying degrees, with women being generally more affected. While thus confirming some previous research findings, some of the present results also called into question other commonly held beliefs about the positive effects of the interest in healthy eating. Namely, healthy orthorexic eating was unrelated to mental health in women but related to better mental health in men. In addition, levels of healthy orthorexia mitigated the negative consequences of orthorexia nervosa. Questions remain whether the nature of this interrelation between healthy orthorexia and orthorexia nervosa as well as gender differences are due to socio-cultural and developmental influences, a combination of both, or due to completely different factors. A more detailed understanding of dimension-specific and gender-specific variables in orthorexic eating will move strategies for prevention and treatment forward.

## Figures and Tables

**Figure 1 nutrients-12-03662-f001:**
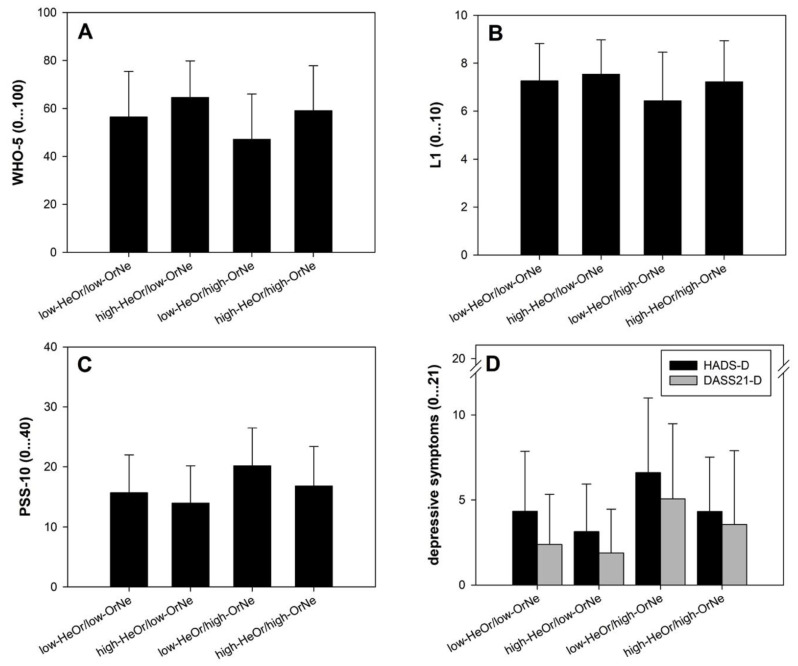
Illustration of the HeOr × OrNe interaction based on median split. HeOr, Healthy orthorexia; OrNe, Orthorexia nervosa; (**A**) WHO-5, Well-being Index; (**B**) L1, Life satisfaction Scale; (**C**) PSS-10, Perceived Stress Scale; (**D**) HADS, Depression Subscale & DASS21, Depression Subscale.

**Table 1 nutrients-12-03662-t001:** Demographic characteristics of the study sample.

Variable	Total Sample (*n* = 783)
Age, years	28.89 ± 11.99 (16–82)
BMI, kg/m2	22.95 ± 4.15 (14.66–46.28)
BMI category, *n* (%)	
Underweight	53 (6.8)
Normal weight	565 (72.2)
Overweight	119 (15.2)
Adiposity	46 (5.9)
Education, *n* (%)	
None, still in school	9 (1.1)
Primary	4 (0.5)
Secondary	58 (7.4)
University-entrance	712 (90.9)
Marital status, *n* (%)	
Unmarried	657 (83.9)
Married	126 (16.1)
Eating style, *n* (%)	
Vegan	51 (6.5)
Any form of vegetarian	125 (16.0)
Rare meat	151 (19.3)
No restrictions	456 (58.2)

BMI, Body Mass Index.

**Table 2 nutrients-12-03662-t002:** Descriptives of the study variables.

Variable	Cronbach’s α	Total Sample (*n* = 783)	Women (*n* = 575)	Men (*n* = 208)
TOS-HeOr	0.84	11.89 ± 5.00	12.08 ± 4.87	11.39 ± 5.30
TOS-ON	0.87	3.36 ± 4.01	3.50 ± 4.23	2.97 ± 3.29
WHO-5 0…100	0.85	57.02 ± 10.01	55.75 ± 18.58	60.52 ± 19.80 **
L1 0…10	--	7.16 ± 1.70	7.11 ± 1.66	7.32 ± 1.79
PSS-10 0…40	0.87	16.43 ± 6.62	17.22 ± 6.42	14.23 ± 6.69 ***
**Study 1**		**Total Sample (*n* = 385)**	**Women** **(*n* = 310)**	**Men** **(*n* = 75)**
HADS-A 0…20	0.79	7.15 ± 3.75	7.26 ± 3.69	6.67 ± 3.96
HADS-D 0…20	0.83	4.49 ± 3.62	4.32 ± 3.36	5.23 ± 4.50
**Study 2**		**Total Sample (*n* = 398)**	**Women** **(*n* = 265)**	**Men** **(*n* = 133)**
DASS21-D	0.89	3.07 ± 3.74	3.33 ± 3.90	2.55 ± 3.37 *
DASS21-A	0.78	1.76 ± 2.60	2.01 ± 2.83	1.27 ± 1.99 **
DASS21-S	0.88	4.47 ± 4.11	5.05 ± 4.20	3.31 ± 3.67 ***

TOS, Teruel Orthorexia Scale; HeOr, Healthy orthorexia; OrNe, Orthorexia nervosa; WHO-5, Well-being Index; L1, Life satisfaction Scale; PSS-10, Perceived Stress Scale; HADS, Hospital Anxiety (A) and Depression (D) Scale; DASS21, Depression (D), Anxiety (A) and Stress (S) Scale. *** *p* < 0.001, ** *p* < 0.01, * *p* < 0.05.

**Table 3 nutrients-12-03662-t003:** Associations between orthorexic eating and mental health measures.

	TOS-HeOr			TOS-OrNe		
Wellbeing or Strain Measure	Total Sample (N = 783)	Women (*n* = 575)	Men (*n* = 208)	Total Sample (N = 783)	Women (*n* = 575)	Men (*n* = 208)
WHO-5	0.205 ***	0.176 ***	0.305 ***^#^	−0.158 ***	−0.191 ***	−0.030
L1	0.101 **	0.045	0.244 ***	−0.203 ***	−0.269 ***	0.018
PSS−10	−0.064	−0.021	−0.217 **	0.287 ***	0.327 ***	0.134
**Study 1**	**Total Sample (N = 385)**	**Women** **(*n* = 310)**	**Men** **(*n* = 75)**	**Total Sample (N = 385)**	**Women** **(*n* = 310)**	**Men** **(*n* = 75)**
HADS-A	0.024	0.082	−0.203	0.384 ***	0.433 ***	0.131
HADS-D	−0.146 **	−0.122 *	−0.191	0.227 ***	0.288 ***	0.041
**Study 2**	**Total Sample (N = 398)**	**Women** **(*n* = 265)**	**Men** **(*n* = 133)**	**Total Sample (N = 398)**	**Women** **(*n* = 265)**	**Men** **(*n* = 133)**
DASS21-D	−0.029	0.061	−0.245 **	0.341 ***	0.414 ***	0.129
DASS21-A	0.040	0.072	−0.075	0.319 ***	0.350 ***	0.202 *
DASS21-S	0.037	0.114	−0.166	0.358 ***	0.432 ***	0.146

TOS, Teruel Orthorexia Scale; HeOr, Healthy orthorexia; OrNe, Orthorexia nervosa; WHO-5, Well-being Index; L1, Life satisfaction Scale; PSS-10, Perceived Stress Scale; HADS, Hospital Anxiety (A) and Depression (D) Scale; DASS21, Depression (D), Anxiety (A) and Stress (S) Scale. Underlined: minimum effect size of r > 0.2. *** *p* < 0.001, ** *p* < 0.01, * *p* < 0.05. ^#^ Significant gender difference.

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
