# Peer review of "The Dark Side of Healthy Eating: Links between Orthorexic Eating and Mental Health"

_nutrients, 2020, doi:10.3390/nu12123662_

Round 1
Reviewer 1 Report
The paper is about the relationship between orthorexia and mental health. The manuscript reports the results from 2 online surveys. I think orthorexia is an interesting topic of research and its relationship with mental health is largely discussed, so I think this manuscript could interest a large audience of researchers and clinicians. I think the author stated clearly that the methodology has several limits, but I have other concerns that should be addressed:
- Do you evaluate the presence of any eating disorder? If yes, how? Looking at data, 7% of the sample is underweight and this could be a sign of anorexia nervosa more than orthorexia.
- In the paper, the author does not take into consideration the relationship between orthorexia and eating disorder. I think this could be something that needs an evaluation.
- When were the surveys conducted (please reported the dates)? The author state that overlapping is unlike, but it is not clear the reason.
- How were distributed the questionnaires in different tests? Are the results confirmed also considering the two studies as separated?
Minor points:
- a 0 is missing at line 157
Reviewer 2 Report
The paper is interesting and the authors have written their results and discussion well. Some minor comments that I have per below in terms of the stats approach.
Section 3.
Is it really important to compare the demographic stats between men vs women? If this doesn't really added value to the manuscript perhaps delete this or add as a footnote.
PCA was carried out, was this mentioned in the stats analysis section? If it is not there yet, please add.
For results interpretation, perhaps it might be worth to identify if the differences were significant or not. The authors have worded it confusingly, for example - with the p = .089 it was written as slightly higher. I'd recommend to write it as 'not significant, but was rated higher' or something along those line to be very clear to the reader. When the p-values are < .05, please also indicate that the differences is significant.
Table 3. Bold to signify gender difference, what does this mean? I can't see the bolds here in the number, does it mean no gender difference? Please amend.
Why was hierarchical multiple regression used? Any specific reason as to normal regressions? It looks like stepwise regression is used here based on line 210?
With the low variance that was explained, what are the authors conclusion here? The fact that for example WHO-5 scale explained 14.6% how can one then conclude from a small variance being explained?
Table 4 & 5 footnotes seems very overkill. Perhaps include as a supplementary table?
